# Mild Left Ventricular Hypertrophy in Middle-Age Male Athletes as a Sign of Masked Arterial Hypertension

**DOI:** 10.3390/ijerph191610038

**Published:** 2022-08-15

**Authors:** Łukasz A. Małek, Agnieszka Jankowska, Lidia Greszata

**Affiliations:** 1Department of Epidemiology, Cardiovascular Disease Prevention and Health Promotion, National Institute of Cardiology, 04-628 Warsaw, Poland; 2National Institute of Cardiology, 04-628 Warsaw, Poland

**Keywords:** differential diagnosis, exaggerated blood pressure response to exercise, ambulatory blood pressure monitoring, exercise test, echocardiography

## Abstract

Mild left ventricular hypertrophy (LVH) has been considered as one of the possible structural, physiological adaptations to regular, intensive physical activity. However, it may also appear as one of the subclinical complications of hypertension. In athletes, the differential diagnosis between these two entities may be complicated as regular physical activity may potentially mask the presence of arterial hypertension. We sought to determine the relation between LVH in middle-age athletes and the presence of hypertension. The study included 71 healthy, male long-time amateur athletes (mean age 41 ± 6 years, 83% endurance and 17% power sports) without known hypertension or any other cardiovascular diseases and with normal self-measured and office blood pressure. All subjects underwent resting electrocardiogram, transthoracic echocardiography, maximal exercise test on a treadmill and ambulatory blood pressure monitoring. LVH was diagnosed as left ventricular wall diameter >11 mm. Hypertension was defined as mean 24 h systolic blood pressure (SBP) ≥ 130 mmHg and/or diastolic blood pressure (DBP) ≥ 80 mmHg. Exaggerated blood pressure response (EBPR) to exercise was defined as SBP ≥ 210 mmHg. LVH (range > 11 to 14 mm) was found in 20 subjects (28%) and hypertension was diagnosed in 33 subjects (46%). Athletes with LVH were more likely to have hypertension than those without LVH (70% vs. 37%, *p* = 0.01). EBPR to exercise was found equally common in athletes with and without LVH (35% vs. 29%, *p* = 0.68), but more often in subjects with hypertension (51% vs. 13%, *p* < 0.001). Presence of LVH and hypertension was equally common in the studied endurance and power sport athletes (*p* = 0.66 and *p* = 0.79, respectively). In comparison to athletes without LVH, those with LVH had larger left atrial size (26 ± 6 vs. 21 ± 4 cm^2^, *p* < 0.001) and a tendency for lower left ventricular diastolic function (E/A 1.2 ± 0.4 vs. 1.5 ± 0.4, *p* = 0.05) and a larger ascending aorta diameter (34 ± 3 vs. 32 ± 3, *p* = 0.05), but a similar left ventricular end-diastolic diameter (51 ± 3 vs. 51 ± 4, *p* = 0.71). The presence of mild left ventricular hypertrophy in middle-age male amateur athletes with normal home and office blood pressure may be considered as a potential sign of masked hypertension. It should not be overlooked as an element of a physiological adaptation to exercise and may warrant further medical evaluation with ambulatory blood pressure monitoring.

## 1. Introduction

Mild, concentric left ventricular hypertrophy (LVH) is considered as one of the possible features of physiological cardiac adaptation to intensive exercise called “athlete’s heart” [1,2]. This is presumed to result from prolonged periods of pressure overload during training and competition. However, LVH is also one of the target organ damage effects of prolonged hypertension [3]. Regular physical activity (PA) has been shown to decrease arterial blood pressure (BP) [4,5]. Aerobic training lasting at least 30 min on most days of the week has been proved to reduce systolic BP (SBP) by around 7 mmHg and diastolic BP (DBP) by 5 mmHg or even more when aerobic training in combination with strength exercises is performed at least 2 to 3 times a week [4,5]. Therefore, regular PA may prevent the onset of hypertension or mitigate hypertension levels. Athletes are not free from hypertension, which has been found in 3.8% of Olympic level athletes from Italy [6]. This number is probably much higher in older, master athletes (above 30–35 years) in whom hypertension develops with age, particularly in amateurs, who often combine training with a not necessarily healthy daily lifestyle. In this group, hypertension may be masked by regular PA, especially in cases of 1st degree hypertension and is often overlooked by the athletes or their personal physicians. Finding of amplitude criteria for LVH in electrocardiogram (ECG) or mild, concentric LVH on transthoracic echocardiogram may often be rationalized as part of physiological adaptation and not lead to additional tests and a more detailed BP assessment [2,7]. Leaving master athletes with undiagnosed hypertension, even mild, puts them at risk of future complications such as aortic dilation, left atrial enlargement with the risk of atrial fibrillation, LVH with fibrosis and the risk of ventricular arrhythmias, diastolic dysfunction of the heart or acceleration of atherosclerosis [4,8]. All these complications may reduce the benefits of regular PA and lead to myocardial infarction, strokes, heart failure, disability and higher mortality in comparison to athletes without or with treated hypertension [4,8]. 

For these reasons, we decided to perform a prospective, one-centre study to determine the relation between LVH in middle-aged athletes with normal home and office blood pressure and the presence of arterial hypertension in order to assess whether finding of LVH may be considered as a potential sign of masked hypertension and not simply as an element of a physiological adaptation to exercise and may warrant further medical evaluation. 

## 2. Materials and Methods

### 2.1. Study Group

This study included 71 healthy volunteers from local sport clubs and societies, who met the following inclusion criteria: (1) male sex, (2) age 30–55 years, (3) documented continuous sport training for at least 5 years with at least 4 trainings a week on average of pure endurance or power sports, (4) several documented BP measurements performed at home in a week prior to study inclusion all with BP < 140/90 mmHg and (5) no chronic diseases, especially no diagnosed hypertension or any other cardiovascular disease. Exclusion criteria were: (1) not meeting inclusion criteria, (2) any chronic medication use including also any performance enhancing substances, (3) obesity (BMI > 30) and (4) lack of signed consent for the participation in the study. 

All study procedures were performed during a single ambulatory visit in the Sports Cardiology Unit of the National Institute of Cardiology in Warsaw between March 2020 and December 2021. Each visit consisted of: medical history and physical examination, resting BP measurement 3 times in a sitting position to exclude initial hypertension (TM-Z/S sphygmomanometer, Tech-Med, Warsaw, Poland), resting ECG (TC 50 Page Writer, Philips, Amsterdam, The Netherlands), transthoracic echocardiography (Vivid S6, General Electric, Boston, MA, USA) and exercise test on a treadmill to maximal exhaustion with Bruce protocol (CardioSoft v.6.5.1, General Electric, Boston, MA, USA). During the exercise test, BP was measured manually at baseline and every 3 min (Big Ben round sphygmomanometer, Riester, Jungingen, Germany). At the end of the testing day, an ambulatory BP monitor (ABPM) was fixed (Schiller BR-102 plus, Baar, Switzerland) for 24 h and all subjects were asked to refrain from exercise in that period. The participants were also asked to refrain from smoking and excessive coffee or energy drink intake 24 h prior to and during testing and to limit exercise intensity to moderate without participating in competitions a few days before the study. Cuff sizes of the BP measurement machines were always adjusted to the arm size. 

Because of the on-going COVID-19 pandemic during the study period we decided to exclude athletes with confirmed COVID-19 infection during 6 months prior to the study procedures or those with any signs or symptoms suggesting a long-COVID disease [9]. 

### 2.2. Definitions

Mild left ventricular hypertrophy (LVH) was diagnosed as left ventricular wall diameter > 11 mm, but not exceeding 15 mm [1,2]. Arterial hypertension was defined as mean 24 h systolic blood pressure (SBP) ≥ 130 mmHg and/or diastolic blood pressure (DBP) ≥ 80 mmHg on ABPM [3]. Exaggerated blood pressure response (EBPR) to exercise was defined as SBP ≥ 210 mmHg [10]. Echocardiographic measurements were performed according to current guidelines [11]. In particular, the relative wall thickness (RWT) was calculated as 2 x posterior wall diameter/end-diastolic LV diameter [12]. An RWT was considered as abnormal if it exceeded 0.42. According to the guidelines of the American Society of Echocardiography, LV geometry in men was classified into 4 groups based on the RWT values and LV mass index (LVMI) as: (1) normal cardiac geometry (RWT ≤ 0.42 and LVMI ≤ 115), (2) concentric remodeling (RWT > 0.42 and LVMI ≤ 115), (3) concentric hypertrophy (RWT > 0.42 and LVMI > 115) and (4) eccentric hypertrophy (RWT ≤ 0.42 and LVMI > 115) [11]. ECG findings were graded based on current international criteria for ECG interpretation in athletes [7].

### 2.3. Clinical Follow-Up 

Clinical follow-up included referral for additional tests due to newly found cardiac conditions. 

### 2.4. Statistical Analysis

All results for categorical variables were presented as a number and percentage. Continuous variables were expressed as median and interquartile range (IQR) or mean and standard deviation (SD) depending on the normality of distribution assessed with means of the Kolmogorov–Smirnov test. Either the chi-square test or the Fisher exact test were used for the comparison of categorical variables, when appropriate. Student’s *t*-test or Mann–Whitney test for unpaired samples was applied to compare two continuous variables depending on the data distribution. All tests were two-sided with a significance level of *p* < 0.05. Statistical analyses were performed with MedCalc statistical software 10.0.2.0 (Ostend, Belgium).

## 3. Results

### 3.1. Baseline Characteristics 

Mean age of the subjects included in the study was 41 ± 6 years. The athletes practised several sport disciplines including running, cycling, rowing as endurance sports (59 subjects—83%) and cross-fit, climbing, weightlifting or boxing as power sports (12 subjects—17%). The mean training load was 7.6 h per week (±1.2 h) divided into a mean of 5.2 days of training (±0.4 days). Athletes were training regularly for a mean of 8.4 years (±1.1 year). 

### 3.2. Relation of LVH to EBPR to Exercise and Hypertension

Mild LVH (range > 11 to 14 mm) was found in 20 subjects (28%) and hypertension was diagnosed in 33 subjects (46%, mean 24 h SBP 132 ± 7 mmHg and DBP 80 ± 5 mmHg). None of the subjects had LVH exceeding 15 mm. Presence of LVH and hypertension was equally common in studied endurance and power sport athletes (*p* = 0.66 and *p* = 0.79, respectively)—Table 1. 

LVH on echocardiography was more likely associated with increase QRS voltage for LVH and with T-wave inversion in infero-lateral leads on resting ECG. 

Athletes with LVH were more likely to have hypertension than those without LVH (70% vs. 37%, *p* = 0.01). EBPR to exercise was found equally common in athletes with and without LVH (35% vs. 29%, *p* = 0.68), but more often in subjects with hypertension (51% vs. 13%, *p* < 0.001). Main findings of the study are presented in Figure 1. 

In comparison to athletes without LVH, those with mild LVH had larger left atrial size (26 ± 6 vs. 21 ± 4 cm^2^, *p* < 0.001) and a tendency for lower left ventricular diastolic function (E/A 1.2 ± 0.4 vs. 1.5 ± 0.4, *p* = 0.05) and a larger ascending aorta diameter (34 ± 3 vs. 32 ± 3, *p* = 0.05), but similar left ventricular end-diastolic diameter (51 ± 3 vs. 51 ± 4, *p* = 0.71)—Figure 2. 

Prevalence of LVH rose from 16% in athletes with normal ABPM and exercise test, to 32% in those with EBPR to exercise and 42% in athletes with newly diagnosed hypertension based on ABPM values (*p* for trend 0.04)—Figure 3. 

### 3.3. Clinical Follow-Up 

All athletes with hypertension were offered life-style management consultation and pharmacological treatment along with periodic monitoring. 

The study revealed a few abnormalities other than hypertension in 11 athletes (15%). Those included the presence of a bicuspid aortic valve without complications in 2 athletes (3%), ascending aorta dilation (48 mm) in 1 athlete without hypertension (1.5%), asymptomatic, significant ST-segment depression during the exercise test in 4 athletes (6%), apical hypertrophic cardiomyopathy based on T-wave inversions in ECG and LV muscle thickness of 14–15 mm in the apical segments and up to 12 mm in other segments with no hypertension in 1 athlete (1.5%), ventricular arrhythmia from the outflow tract in 1 athlete (1.5%), T-wave inversion in inferior and lateral ECG leads in 2 athletes (3%)—which led to cardiac magnetic resonance discovery of prior myocarditis and exercise-induced left bundle branch block (LBBB) in 1 athlete (1.5%) in whom a subsequent computed tomography of coronary arteries (CTCA) excluded coronary artery disease. CTCA was also performed in 2 out of 4 athletes (2 refused) with exercise induced ST-segment depression described above, which turned out to be a false negative in 1 case, but showed coronary artery anomaly (slit like origin of the circumflex artery) in the second case. All of these athletes were also offered further treatment and monitoring as per guidelines. 

## 4. Discussion

We have demonstrated that middle-age athletes who have been engaged for many years in endurance and power sport disciplines and have mild LVH with normal resting self-measured and office blood pressure are more likely to suffer from undiagnosed hypertension than their counterparts with normal left ventricular thickness. In other terms, mild LVH may be considered not only as a marker of physiological adaptation to exercise, but could also be viewed as a risk factor for masked hypertension in that population. This situation may be often overlooked because according to current international guidelines for ECG interpretation in athletes, the finding of increased QRS voltage does not necessarily require further evaluation in asymptomatic athletes with no family history of inherited cardiac disease or sudden cardiac death and normal resting office blood pressure [7]. In our study increased QRS voltage for LVH was found in one fifth of subjects and over half of them were later found to have hypertension. 

An observation that middle-age athletes have masked hypertension, which may be elucidated on ABPM in as many as 38% of cases has been made by Trachsel et al. in their study on 108 normotensive runners [13]. The authors also found that those patients had lower diastolic function, higher LV mass to volume ratio, but no difference in RWT and left atrial volume index. No exercise test was done in that study to assess the co-existence of EBPR response to exercise with LVH and hypertension. A following systematic review demonstrated that the prevalence of hypertension in athletes varied from 0 to 83% and that some studies showed an association between high BP and LVH [14]. The authors noted that the methods in many studies were poorly standardized. In this review it was also demonstrated, unlike in our study, that strength athletes had higher BP than endurance athletes and that BP tended to be higher in those training over 10 h/week compared to others. In our initial analysis on 30 ultra-marathon runners with normal office BP values we were able to demonstrate that left ventricular thickness assessed by cardiac magnetic resonance was gradually increasing from optimal/normal BP, through EBPR to exercise and high normal BP, to the highest values in runners with high normal BP and EBPR to exercise [15]. Those findings and our current report are in line with recent analysis from the Italian group on Olympic athletes where LVH was related to BP at rest and during exercise, larger body weight, body mass index and fat percentage, but was not associated with greater physical performance [16]. 

Therefore, to further examine the relation between intensive sport activity, LVH and hypertension we decided for the current study not only to use ABPM instead of office BP measurements, but also to analyse further the role of EBPR. We have shown that EBPR was more likely to be found in athletes with hypertension and that LVH was more prevalent in athletes with EBPR than in those with normal BP and normal response of BP to exercise. It has been shown earlier that exercise-induced hypertension is related to higher angiotensin II activity and reduced nitric oxide levels in middle-age long-distance runners [17]. This may explain some of the other findings related to EBPR in athletes and in general population. A review of 16 studies in over 23 thousand of subjects with normal resting BP but EBPR to exercise and followed for over 5 years demonstrated that EBPR was a predictor of hypertension development in the future [18]. In a study by Tahir et al. on triathletes, unfavourable blood pressure response to exercise was associated with post-race cardiac dysfunction, which could explain the occurrence of myocardial fibrosis in this group [19]. Finally, EBPR was also shown to be a risk factor of sudden cardiac death in men without a history of cardiovascular diseases [20]. 

From a clinical point of view, high prevalence of masked hypertension in middle-age ambitious male athletes may explain some of the complications characteristic of this population such as high coronary calcium score, high atherosclerotic plaque burden, myocardial fibrosis or higher risk of atrial fibrillation in comparison to general population [8,21]. A previous meta-analysis demonstrated that patients with masked hypertension, despite only mildly elevated BP values, have significantly increased rates of cardiovascular events and all-cause mortality than normotensives [22].

Therefore, we believe that in each case of suspected LVH based on ECG or in LVH found on echocardiography, a rigorous differential diagnosis between hypertensive heart disease and a normal athlete’s heart condition according to criteria described elsewhere may be mandatory even despite normal self-measured or office BP [23,24]. 

Apart from the main findings of our study, it is also worthwhile to mention the relatively high number of accidental findings during screening, as one sixth of the studied subjects presented some abnormalities. This may be considered as a call for action to promote regular pre-participation examination by sports cardiologists in middle-age amateur athletes, who often test their limits in sport competitions. 

Our study has some limitations. First of all, we were able to include mainly endurance athletes with a much lower number of power athletes. This was related to lower recruitment of power athletes due to our exclusion criteria, mainly illicit substances use. Secondly, we decided to limit our observations to cardiac complications of hypertension and did not analyse other target organ damage, cardiac fibrosis and prospective events. Nevertheless, we believe that our study gives robust evidence for the relation between LVH, EBPR to exercise, hypertension and potential heart complications such as lower diastolic function and a tendency for higher left atrial and ascending aorta size. Finally, we decided not to exclude two patients with signs of left anterior hemiblock (LAH) from the analysis. International ECG criteria do not specifically refer to that entity in athletes [7]; both of the patients with LAH apart from mild LVH did not have any other abnormalities on echocardiogram and had a normal exercise test. Both of the patients were found to have masked hypertension. Therefore, we decided to consider LAH as a sign of delayed conductance related to target organ damage in the form of LVH due to newly diagnosed hypertension in these patients. 

## 5. Conclusions

The presence of mild left ventricular hypertrophy in middle-age male amateur athletes with normal home and office blood pressure may be considered as a potential sign of masked hypertension and not overlooked as an element of a physiological adaptation to exercise. It may warrant further medical evaluation with ambulatory blood pressure monitoring.

## Figures and Tables

**Figure 1 ijerph-19-10038-f001:**
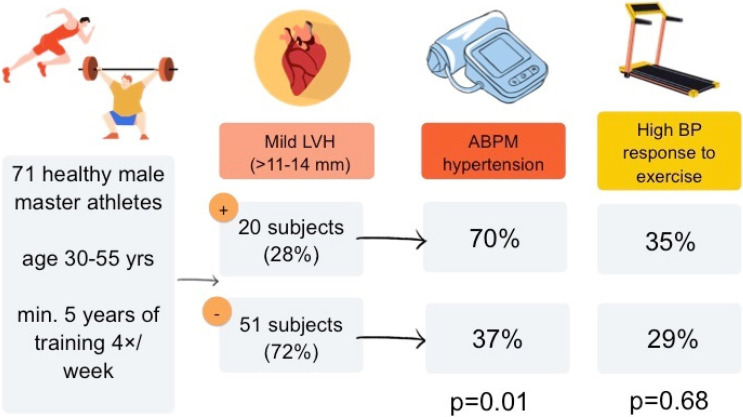
Graphical abstract presenting main findings of the study. ABPM—24 h ambulatory blood pressure monitoring, BP—blood pressure, LVH—left ventricular hypertrophy.

**Figure 2 ijerph-19-10038-f002:**
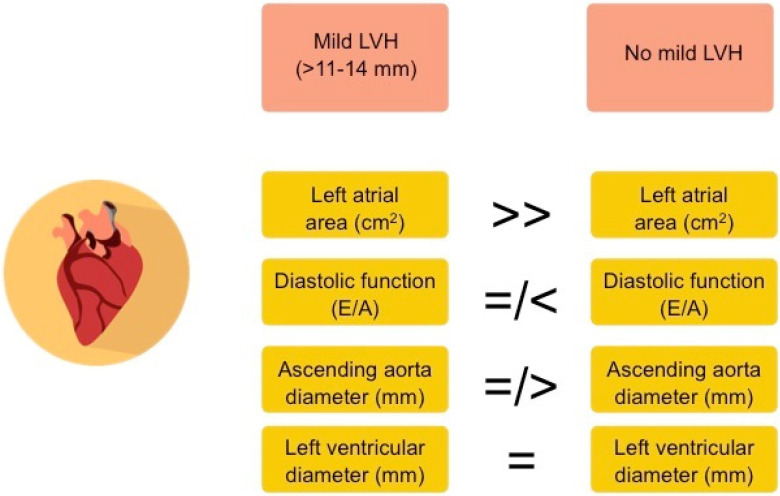
Main differences in echocardiographic parameters between studied athletes with and without left ventricular hypertrophy (LVH).

**Figure 3 ijerph-19-10038-f003:**
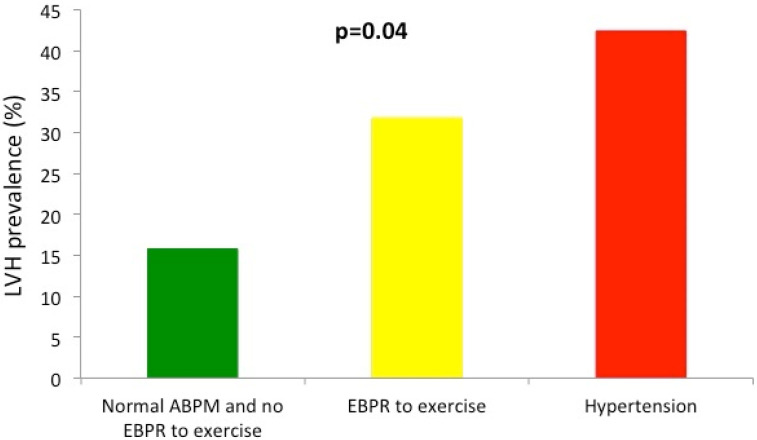
Prevalence of LVH in the studied group in athletes with normal ambulatory blood pressure monitoring (ABPM) values and not exaggerated blood pressure response (EBPR) to exercise (green bar), in athletes with EBPR to exercise (yellow bar) and in those with diagnosed hypertension based on ABPM values (red bar). *p* for trend = 0.04.

**Table 1 ijerph-19-10038-t001:** Baseline characteristics and test results in subjects with and without mild left ventricular hypertrophy.

Parameter	Mild Left Ventricular Hypertrophy (+) n = 20 (28%)	Mild Left Ventricular Hypertrophy (−) n = 51 (72%)	*p*-Value
Baseline characteristics			
Age (years, SD)	42 ± 6	41 ± 6	0.62
Weight (kg, SD)	82 ± 13	79 ± 11	0.32
Height (cm, SD)	180 ± 6	180 ± 7	0.96
BMI (SD)	25 ± 3	24 ± 3	0.27
Sports discipline (n, %) endurance power	16 (80) 4 (20)	43 (84) 8 (16)	0.66
Weekly training load (h, SD)	8 (1)	7 (2)	0.56
Time of training (years, SD)	8 (2)	8 (1)	0.61
ECG parameters			
Bradycardia (n, %)	11 (55)	26 (51)	0.76
Left axis deviation	1 (5)	1 (2)	0.90
Increase QRS voltage for LVH	7 (35)	7 (14)	0.04
Left atrial enlargement	8 (40)	13 (25)	0.23
Left anterior hemiblock	2 (10)	0 (0)	0.08
T-wave inversion in infero-lateral leads	3 (15)	0 (0)	0.02
TTE parameters			
LVEDd (mm, SD)	51 ± 3	51 ± 4	0.71
IVSd (mm, SD)	12 ± 1	9 ± 1	<0.001
PWd (mm, SD)	12 ± 1	9 ± 1	<0.001
E/A (SD)	1.2 ± 0.4	1.5 ± 0.4	0.05
E/e’ (SD)	7 ± 2	6 ± 1	0.26
LAd (mm, SD)	36 ± 21	36 ± 20	0.96
LAA (cm^2^, SD)	26 ± 6	21 ± 4	<0.001
AAd (mm, SD)	34 ± 3	32 ± 3	0.05
RWT (cm^2^)	0.46 ± 0.05	0.38 ± 0.03	<0.001
LVMI (g/ m^2^)	121 ± 18	94 ± 17	<0.001
LV geometry (n, %)			<0.001
Normal geometry	1 (5)	39 (76)	
Concentric remodelling	7 (35)	7 (14)	
Concentric hypertrophy	8 (40)	0 (0)	
Eccentric hypertrophy	4 (20)	5 (10)	
ABPM parameters			
Mean daily SBP (mmHg, SD)	130 ± 12	122 ± 9	<0.001
Mean daily DBP (mmHg, SD)	78 ± 7	74 ± 6	0.03
Hypertension (n, %)	14 (70)	19 (37)	0.01
Exercise test parameters			
BP at maximal exertion (mmHg, SD)	191 ± 29	191 ± 25	0.92
EBPR to exercise (n, %)	7 (35)	15 (29)	0.68

AAd—ascending aorta diameter, BMI—body mass index, BP—blood pressure, DBP—diastolic blood pressure, E/A—early to late mitral inflow velocity, E/e’—early mitral inflow velocity and mitral annular early diastolic velocity, EBPR—exaggerated blood pressure response, LVEDd—end-diastolic diameter of the left ventricle, IVSd—interventricular septal diameter, LAd—left atrial diameter, LAA—left atrial area, LV—left ventricle, LVMI—left ventricular mass index, PWd—posterior wall diameter, RWT—relative wall thickness, SBP—systolic blood pressure, SD—standard deviation.

## Data Availability

Data are available on request from the authors.

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
