# Peer review of "Mild Left Ventricular Hypertrophy in Middle-Age Male Athletes as a Sign of Masked Arterial Hypertension"

_ijerph, 2022, doi:10.3390/ijerph191610038_

Round 1
Reviewer 1 Report
Thanks for the opportunity to review the manuscript. in which the relationship between benign left ventricular hypertrophy and the occurrence of arterial hypertension in middle-aged athletes was sought.
1. Adding sections to the abstract will make it clearer
2. The introduction should be supplemented with the current state of knowledge on the topic under study
3. The research methodology is clearly and properly described. The authors provide criteria for inclusion and exclusion from the study. Has the sample size been calculated somehow? if so worth mentioning.
4. Statistic analysis, the description shows that the data were characterized by a normal distribution and deviating from it. Therefore, it seems logical to confine the results to the decimal logarithm in order to adjust them to meet the normal distribution, and the effect size can also be given. Please take this as a suggestion.
5. General and conclusions are written appropriate
Author Response
Thanks for the opportunity to review the manuscript. in which the relationship between benign left ventricular hypertrophy and the occurrence of arterial hypertension in middle-aged athletes was sought.
1. Adding sections to the abstract will make it clearer
We have followed the policies of the journal, where a single paragraph is advised without subdivision into typical sections – objectives, methods, results and conclusions. However, all of those parts are covered in the abstract.
2. The introduction should be supplemented with the current state of knowledge on the topic under study
There are no previous studies addressing the topic of the study in the way it was done in our study. Other information regarding current state of knowledge on the topic of physiological vs. pathological adaptation of the heart to exercise has been highlighted in the introduction section. We also discuss the other papers addressing the issue in the discussion section. Therefore we would like to avoid redundancy.
3. The research methodology is clearly and properly described. The authors provide criteria for inclusion and exclusion from the study. Has the sample size been calculated somehow? if so worth mentioning.
The power calculations were not done beforehand. However, we have found an almost double difference in the prevalence of hidden hypertension between athletes with and without LVH, which gives a very high power of the results.
4. Statistic analysis, the description shows that the data were characterized by a normal distribution and deviating from it. Therefore, it seems logical to confine the results to the decimal logarithm in order to adjust them to meet the normal distribution, and the effect size can also be given. Please take this as a suggestion.
All of our quantitative data had a normal distribution so we have used mean plus standard deviation. We did not have any data deviating from normality. The normality was assed with the use of Kolmogorov-Smirnov test. For that reason we did not have to use median and interquartile range or decimal logarithm in order to adjust the results to meet the normal distribution.
5. General and conclusions are written appropriate
Thank you
Reviewer 2 Report
Lukasz and co-workers report a clinical study on middle-age male athletes to determine the relation between left ventricle hypertrophy and hypertension. The topic is relevant to better understand the association between these two conditions and to early identify signs of hypertension thus contributing to prevent cardiovascular diseases.
The manuscript is very well written and very easy to follow, with scientific support from the literature and logic engagement of the results/discussion.
I have a very simple misunderstanding concerning line 137; it is not clear to what participants refers the mean SBP and DBP presented. It cannot be the mean of the 33 classified as having hypertension…
Overall, my only criticism is on the BP levels referred, which although showing statistical significance, are quite low. As so, the risk for these levels of BP and future cardiovascular disease/events should be evidenced with literature support. Otherwise, the study maintains its interest but the discussion/conclusion should be completely reformulated.
Author Response
Lukasz and co-workers report a clinical study on middle-age male athletes to determine the relation between left ventricle hypertrophy and hypertension. The topic is relevant to better understand the association between these two conditions and to early identify signs of hypertension thus contributing to prevent cardiovascular diseases.
Malek and co-workers :)
The manuscript is very well written and very easy to follow, with scientific support from the literature and logic engagement of the results/discussion.
I have a very simple misunderstanding concerning line 137; it is not clear to what participants refers the mean SBP and DBP presented. It cannot be the mean of the 33 classified as having hypertension…
In fact, these are the mean SBP and DBP values in athletes with hypertension (33 out of 77). It was only mildly increased as occurs in subjects with masked hypertension. The mean values in athletes with and without LVH are reported in Table 1. The higher values would be probably easy detectable with home BP measurements or office BP measurements.
Overall, my only criticism is on the BP levels referred, which although showing statistical significance, are quite low. As so, the risk for these levels of BP and future cardiovascular disease/events should be evidenced with literature support. Otherwise, the study maintains its interest but the discussion/conclusion should be completely reformulated.
We have expected that mean BP values in athletes with masked may be relatively low, otherwise hypertension would be detected earlier and not masked. However, it has been demonstrated that even these relatively mildly elevated BP levels can lead to higher risk of target organ damage and macroscopic complications (Palla M et al. 2018). We have added the following sentence with reference to the study of Palla et al. in the discussion section: “A previous meta-analysis demonstrated that patients with masked hypertension, despite only mildly elevated BP values, have significantly increased rates of cardiovascular events and all-cause mortality than normotensives [22].”
For this reason we believe, as stated in the discussion, that athletes with masked hypertension (even mild) are at higher risk of aortic complications, atrial fibrillation, progression towards heart failure and potentially coronary artery disease (although not assessed in our study, but reported elsewhere [unpublished data] – https://www.medscape.co.uk/viewarticle/male-endurance-athletes-often-have-undiagnosed-hypertension-2021a1001vyt
Reviewer 3 Report
Thank you for providing me the opportunity to review the paper entitled "Mild left ventricular hypertrophy in middle-age male athletes as a sign of masked arterial hypertension" by Brugger et al.
General comments:
The study investigated to determine the relation between left ventricular hypertrophy (LVH) in middle-age athletes and the presence of hypertension. The authors concluded that The presence of mild left ventricular hypertrophy in middle-age male amateur athletes should always be considered as a potential sign of masked hypertension and not overlooked as an element of a physiological adaptation to exercise and should always warrant further rigorous medical evaluation. It is a well-conducted study, and the manuscript itself is well written and the findings are well presented. Furthermore, the authors used new technique to determination of LA pump function in the present study. I have some comments.
Major:
1. the authors used the definition of mild LVH that was diagnosed as left ventricular wall diameter >11mm, but not exceeding 15mm. however, practically, that definition was not mild LVH but also just LVH, which means that the patients with LVH >11mm have target organ damage associated with hypertension or other risk factors and those patients are not healthy candidates but also may have hypertension already. So please explain why the author thought that the patients with LVH can be healthy candidates in this study.
2, In Table 1, in mild LVH group, 2 of those candidates have Left anterior hemiblock. I think these patients must be excluded. Left anterior hemiblock is not normal variant, and healthy atheletes can not have those arrhythmia. Please explain.
3, In Table 1, in ABPM parameters, some of enrolled candidates had already hypertension; definition -wise.--> in mild LVH group, some candidates had mean daily SBP/DBP 142/85mmHg, in non-LVH group, some of them had mean daily SBP/DBP 131/80mmHg. Please exclude those patients with hypertension.
Author Response
Thank you for providing me the opportunity to review the paper entitled "Mild left ventricular hypertrophy in middle-age male athletes as a sign of masked arterial hypertension" by Brugger et al.
Malek et al J
General comments:
The study investigated to determine the relation between left ventricular hypertrophy (LVH) in middle-age athletes and the presence of hypertension. The authors concluded that The presence of mild left ventricular hypertrophy in middle-age male amateur athletes should always be considered as a potential sign of masked hypertension and not overlooked as an element of a physiological adaptation to exercise and should always warrant further rigorous medical evaluation. It is a well-conducted study, and the manuscript itself is well written and the findings are well presented. Furthermore, the authors used new technique to determination of LA pump function in the present study. I have some comments.
Major:
- the authors used the definition of mild LVH that was diagnosed as left ventricular wall diameter >11mm, but not exceeding 15mm. however, practically, that definition was not mild LVH but also just LVH, which means that the patients with LVH >11mm have target organ damage associated with hypertension or other risk factors and those patients are not healthy candidates but also may have hypertension already. So please explain why the author thought that the patients with LVH can be healthy candidates in this study.
Mild LVH in athletes, as stated in the introduction section, does not necessarily need to be associated with hypertension and a priori defined as target organ damage. In fact, as demonstrated in milestone studies by Marganroth et al. and followed by studies by Pellicia A et al. and Basavarajaiah S et al. LVH can be found in around 15% of white athletes in response to chronic, intensive exercise. It is believed to be caused by periods of increased pressure overload during exercise rather than hypertension. Therefore we wanted to assess of how many of the athletes with LVH may have hidden hypertension despite normal BP home or ambulatory values.
2, In Table 1, in mild LVH group, 2 of those candidates have Left anterior hemiblock. I think these patients must be excluded. Left anterior hemiblock is not normal variant, and healthy atheletes can not have those arrhythmia. Please explain.
Left anterior hemiblock (LAH) is not referenced in the international criteria of ECG interpretation in athletes (Sharma S et al. 2017) and its prevalence in this group is largely unknown. Definitely it currently cannot be considered as necessarily abnormal in athletes. The prevalence of LAH in the general population is between 1% and 6% (3% in our study). For these reason we have decided to keep those athletes in the study especially as it was not associated with signs of coronary artery disease in echocardiogram or positive exercise test. In fact it might have been in our situation a marker of delayed conductance due to LVH as it was found only in the subgroup of athletes with LVH.
3, In Table 1, in ABPM parameters, some of enrolled candidates had already hypertension; definition -wise.--> in mild LVH group, some candidates had mean daily SBP/DBP 142/85mmHg, in non-LVH group, some of them had mean daily SBP/DBP 131/80mmHg. Please exclude those patients with hypertension.
All of the included patients as stated in the inclusion criteria had to have normal self-assessed and ambulatory blood pressure so they could not be classified as having hypertension. Only by the use of ABPM we have found out that they had hypertension, which was the purpose of the study – to see if mild LVH can be a marker of masked hypertension in that group despite normal office or home measurements of blood pressure. And we have found exactly that. Hypertension defined by ABPM was found in both groups, but more likely in the LVH group.
Round 2
Reviewer 3 Report
Major:
- the authors used the definition of mild LVH that was diagnosed as left ventricular wall diameter >11mm, but not exceeding 15mm. however, practically, that definition was not mild LVH but also just LVH, which means that the patients with LVH >11mm have target organ damage associated with hypertension or other risk factors and those patients are not healthy candidates but also may have hypertension already. So please explain why the author thought that the patients with LVH can be healthy candidates in this study.
Mild LVH in athletes, as stated in the introduction section, does not necessarily need to be associated with hypertension and a priori defined as target organ damage. In fact, as demonstrated in milestone studies by Marganroth et al. and followed by studies by Pellicia A et al. and Basavarajaiah S et al. LVH can be found in around 15% of white athletes in response to chronic, intensive exercise. It is believed to be caused by periods of increased pressure overload during exercise rather than hypertension. Therefore we wanted to assess of how many of the athletes with LVH may have hidden hypertension despite normal BP home or ambulatory values.
--> the authors used references Marganroth et al. Pellicia A et al. and Basavarajaiah S et al. I agree these paper can be milestone studies. However I wonder if the authors understand what those paper's message are. "LVH can be found in around 15% of white athletes in response to chronic, intensive exercise. It is believed to be caused by periods of increased pressure overload during exercise rather than hypertension." --> this means LVH is hidden target organ damage, caused by periods of increased pressure overload during exercise. this increased pressure overload is " hypertension". please review again about hypertension!!
2, In Table 1, in mild LVH group, 2 of those candidates have Left anterior hemiblock. I think these patients must be excluded. Left anterior hemiblock is not normal variant, and healthy atheletes can not have those arrhythmia. Please explain.
Left anterior hemiblock (LAH) is not referenced in the international criteria of ECG interpretation in athletes (Sharma S et al. 2017) and its prevalence in this group is largely unknown. Definitely it currently cannot be considered as necessarily abnormal in athletes. The prevalence of LAH in the general population is between 1% and 6% (3% in our study). For these reason we have decided to keep those athletes in the study especially as it was not associated with signs of coronary artery disease in echocardiogram or positive exercise test. In fact it might have been in our situation a marker of delayed conductance due to LVH as it was found only in the subgroup of athletes with LVH.
--> i means if the authors really want to use those data about mild LVH in heathy athletes. i recommend to excluded those LAH. definition wise, LAH is not normal....!!
3, In Table 1, in ABPM parameters, some of enrolled candidates had already hypertension; definition -wise.--> in mild LVH group, some candidates had mean daily SBP/DBP 142/85mmHg, in non-LVH group, some of them had mean daily SBP/DBP 131/80mmHg. Please exclude those patients with hypertension.
All of the included patients as stated in the inclusion criteria had to have normal self-assessed and ambulatory blood pressure so they could not be classified as having hypertension. Only by the use of ABPM we have found out that they had hypertension, which was the purpose of the study – to see if mild LVH can be a marker of masked hypertension in that group despite normal office or home measurements of blood pressure. And we have found exactly that. Hypertension defined by ABPM was found in both groups, but more likely in the LVH group.
--> i agree the author's response. yeah, if the authors wanted to describe LVH (not mild LVH) with masked hypertension, how about describing those as follows :?
" we evaluated the healthy athletes without previous history of hypertension. those healthy candidates had normal office & home BP. however, in this study, we found that the candidates with LVH had hypertension or pre-hypertension using ABPM. therefore, LVH without any symptoms and normal office/home BP can be good predictor for masked hypertension"
Author Response
Major:
- the authors used the definition of mild LVH that was diagnosed as left ventricular wall diameter >11mm, but not exceeding 15mm. however, practically, that definition was not mild LVH but also just LVH, which means that the patients with LVH >11mm have target organ damage associated with hypertension or other risk factors and those patients are not healthy candidates but also may have hypertension already. So please explain why the author thought that the patients with LVH can be healthy candidates in this study.
Mild LVH in athletes, as stated in the introduction section, does not necessarily need to be associated with hypertension and a priori defined as target organ damage. In fact, as demonstrated in milestone studies by Marganroth et al. and followed by studies by Pellicia A et al. and Basavarajaiah S et al. LVH can be found in around 15% of white athletes in response to chronic, intensive exercise. It is believed to be caused by periods of increased pressure overload during exercise rather than hypertension. Therefore we wanted to assess of how many of the athletes with LVH may have hidden hypertension despite normal BP home or ambulatory values.
- the authors used references Marganroth et al. Pellicia A et al. and Basavarajaiah S et al. I agree these paper can be milestone studies. However I wonder if the authors understand what those paper's message are. "LVH can be found in around 15% of white athletes in response to chronic, intensive exercise. It is believed to be caused by periods of increased pressure overload during exercise rather than hypertension." --> this means LVH is hidden target organ damage, caused by periods of increased pressure overload during exercise. this increased pressure overload is " hypertension". please review again about hypertension!!
Mild LVH in athletes is not considered as target organ damage as it was never shown to lead to adverse outcomes. In general athletes live longer, are free from diastolic dysfunction related to LVH (in fact their diastolic function of the LV is often supranormal). On the contrary, patients with LVH due to hypertension indeed have target organ damage as in this case LVH was related to worse outcomes, worse diastolic function, ventricular arrhythmia etc. In athletes mild LVH is a form of physiological adaptation to meet hemodynamic demands of the prolonged, intensive physical activity with healthy hypertrophy of the cardiomyocytes. In hypertension there is, apart of hypertrophy, also marked share of myocardial microfibrosis and therefore it is considered as target organ damage. In the same way, we do not consider balanced enlargement of heart chambers in athletes, including enlargement of the left ventricle as target organ damage or dilated cardiomyopathy.
2, In Table 1, in mild LVH group, 2 of those candidates have Left anterior hemiblock. I think these patients must be excluded. Left anterior hemiblock is not normal variant, and healthy atheletes can not have those arrhythmia. Please explain.
Left anterior hemiblock (LAH) is not referenced in the international criteria of ECG interpretation in athletes (Sharma S et al. 2017) and its prevalence in this group is largely unknown. Definitely it currently cannot be considered as necessarily abnormal in athletes. The prevalence of LAH in the general population is between 1% and 6% (3% in our study). For these reason we have decided to keep those athletes in the study especially as it was not associated with signs of coronary artery disease in echocardiogram or positive exercise test. In fact it might have been in our situation a marker of delayed conductance due to LVH as it was found only in the subgroup of athletes with LVH.
- i means if the authors really want to use those data about mild LVH in heathy athletes. i recommend to excluded those LAH. definition wise, LAH is not normal....!!
As stated earlier we do not find a justrified reason to exclude those patients from the analysis as explained in the initial response. It would also lower the number of analyzed cases. However, following your comment and Editor suggestion we have included the following statement in the limitations section: „Finally, we have decided not to exclude two patients with signs of left anterior hemiblock (LAH) from the analysis. International ECG criteria do not specifically refer to that entity in athletes [7], both of the patients with LAH apart of mild LVH did not have any other abnormalities on echocardiogram and had a normal exercise test. Both of the patients were found to have masked hypertension. Therefore we decided to consider LAH as a sign of delayed conductance related to target organ damage in form of LVH due to newly diagnosed hypertension in these patients.”
3, In Table 1, in ABPM parameters, some of enrolled candidates had already hypertension; definition -wise.--> in mild LVH group, some candidates had mean daily SBP/DBP 142/85mmHg, in non-LVH group, some of them had mean daily SBP/DBP 131/80mmHg. Please exclude those patients with hypertension.
All of the included patients as stated in the inclusion criteria had to have normal self-assessed and ambulatory blood pressure so they could not be classified as having hypertension. Only by the use of ABPM we have found out that they had hypertension, which was the purpose of the study – to see if mild LVH can be a marker of masked hypertension in that group despite normal office or home measurements of blood pressure. And we have found exactly that. Hypertension defined by ABPM was found in both groups, but more likely in the LVH group.
--> i agree the author's response. yeah, if the authors wanted to describe LVH (not mild LVH) with masked hypertension, how about describing those as follows :?
" we evaluated the healthy athletes without previous history of hypertension. those healthy candidates had normal office & home BP. however, in this study, we found that the candidates with LVH had hypertension or pre-hypertension using ABPM. therefore, LVH without any symptoms and normal office/home BP can be good predictor for masked hypertension"
Thank you for that comment. We have included this suggested additional information in the reformatted conclusion (both in abstract and at the end of the manuscript): „The presence of mild left ventricular hypertrophy in middle-age male amateur athletes with normal office and home blood pressure may be considered as a potential sign of masked hypertension and not overlooked as an element of a physiological adaptation to exercise and may warrant further medical evaluation with ambulatory blood pressure monitoring.”